# Evaluation of the Safety and Immunogenicity of Fractional Intradermal COVID-19 Vaccines as a Booster: A Pilot Study

**DOI:** 10.3390/vaccines10091497

**Published:** 2022-09-08

**Authors:** Suvimol Niyomnaitham, Somruedee Chatsiricharoenkul, Zheng Quan Toh, Sansnee Senawong, Chatkamol Pheerapanyawaranun, Supaporn Phumiamorn, Paul V. Licciardi, Kulkanya Chokephaibulkit

**Affiliations:** 1Siriraj Institute of Clinical Research, Faculty of Medicine Siriraj Hospital, Mahidol University, Bangkok 10700, Thailand; 2Department of Pharmacology, Faculty of Medicine Siriraj Hospital, Mahidol University, Bangkok 10700, Thailand; 3Murdoch Children’s Research Institute, Parkville, Melbourne, VIC 3052, Australia; 4Department of Paediatrics, The University of Melbourne, Parkville, Melbourne, VIC 3010, Australia; 5Department of Immunology, Faculty of Medicine Siriraj Hospital, Mahidol University, Bangkok 10700, Thailand; 6Department of Medical Sciences, Ministry of Public Health, Nonthaburi 11000, Thailand; 7Department of Pediatrics, Faculty of Medicine Siriraj Hospital, Mahidol University, Bangkok 10700, Thailand

**Keywords:** COVID-19 vaccination, intradermal, heterologous, booster, Thailand

## Abstract

Intradermal vaccination using fractional dosages of the standard vaccine dose is one strategy to improve access to COVID-19 immunization. We conducted a pilot study in healthy adults in Thailand to evaluate the safety and immunogenicity of intradermal administration of fractional doses of ChAdOx1 (1/5th of standard dosage) or BNT162b2 (1/6th of standard dosage) to individuals previously vaccinated (prime) with two-dose intramuscular CoronaVac, ChAdOx1 or BNT162b2. Following an initial immunogenicity exploratory phase for each vaccine combination group (*n* = 10), a total of 135 participants (*n* = 45 per group) were recruited to 3 groups (CoronaVac prime-intradermal BNT162b2 boost, CoronaVac prime-intradermal ChAdOx1 boost and ChAdOx1 prime-intradermal BNT162b2 boost) and their immunogenicity data were compared to a previous cohort who received the same vaccine intramuscularly. Two weeks following booster vaccination, neutralizing antibodies against the delta variant were similar between the participants who received intradermal and intramuscular vaccination. However, neutralizing antibodies against the omicron variant in the intradermal BNT162b2 boost groups were ~6-fold lower, while the levels in the ChAdOx1 boost group were similar compared to their respective vaccine regimen given intramuscularly. The intradermal booster significantly increased spike-specific T cell responses in all three groups from pre-booster levels. Local and systemic adverse reactions were milder in intradermal compared to intramuscular injections. Further studies are needed to evaluate the clinical relevance of these findings and the feasibility of administration of intradermal COVID-19 vaccines.

## 1. Introduction

As of June 2022, eleven COVID-19 vaccines (four non-replicating viral vector, three inactivated, two RNA, two protein subunit) have been granted Emergency Use Listing by the World Health Organization (WHO) [1]. These vaccines given as two- or one-dose schedules (Ad26.CoV.S, Johnson and Johnson) were highly effective against severe COVID-19 and deaths, and also provided some protection against SARS-CoV-2 infection prior to the omicron variant [2]. However, a booster or third vaccine dose is needed to protect against the SARS-CoV-2 omicron variant, due to its ability to evade vaccine-induced immunity [3]. Neutralizing antibodies are thought to be the primary mechanism of protection, but the concentration needed has not been identified [4]. Furthermore, T cell immunity is increasingly recognized as an important marker of protection against severe COVID-19, and is conserved across multiple SARS-CoV-2 variants [5].

In high-income countries, most of the eligible population have received two or three doses of COVID-19 vaccines, but only 20% in LMICs have received one dose, due to inequitable distribution of COVID-19 vaccines [6]. A few strategies have been proposed to improve access to COVID-19 vaccination, which include fractional dosing, intradermal vaccination, heterologous vaccine regimens and dose stretching [7,8,9,10,11,12]. While heterologous regimens and dose stretching have been studied extensively, few studies have examined intradermal COVID-19 vaccination. The dermis consists of multiple antigen presenting cells that improve vaccine immunogenicity [13]. As such, intradermal vaccination typically requires a lower vaccine dose than intramuscular vaccination, hence maximizing the use of available vaccines. While the reactogenicity following intradermal vaccination was generally higher for localized reactions, systemic reactions were lower compared to intramuscular vaccination, as observed in influenza vaccination [14,15]. Intradermal administration may increase vaccine uptake among those with vaccine hesitancy due to safety concerns of standard intramuscular injection. Intradermal vaccination is already in routine use for bacillus Calmette–Guérin and rabies vaccines [16].

Since early 2021, Thailand has mainly relied on the chimpanzee adenovirus-vector (ChAdOx1) vaccine (AZD1222, *AstraZeneca/Oxford*) and the whole-cell inactivated vaccines, such as CoronaVac (*Sinovac, Life Sciences*), for protection against COVID-19. Subsequently, a limited amount of BNT162b2 vaccines (*Pfizer–BioNTech*) were available. One way to maximize the use of these vaccines is intradermal vaccination using lower vaccine dosage, but whether similar immunogenicity can be achieved as with standard intramuscular vaccination is unclear. Heterologous vaccine regimens were recommended by the WHO under certain circumstances related to vaccine supply and access [15]. This study aims to assess the safety and immunogenicity of intradermal administration of heterologous or homologous booster vaccination using fractional BNT162b2 (1/6th of standard dose) or ChAdOx1 (1/5th of standard dose) in individuals who have completed the standard intramuscular primary series vaccination of either ChAdOx1, CoronaVac or BNT162b2. These dosages are typical for intradermal vaccination [14,16,17].

## 2. Materials and Methods

### 2.1. Study Design and Participants

This is a prospective and open-label study conducted at the Faculty of Medicine Siriraj Hospital, Mahidol University, Bangkok, Thailand, between September 2021 and February 2022. Participants were healthy adults aged 18 years or older and have received homologous two-dose primary series intramuscular vaccination of either CoronaVac (CoronaVac-prime), ChAdOx1 (ChAdOx1-prime), or BNT162b2 (BNT162b2-prime) 4–12 weeks prior to recruitment. The participants were excluded from the study if they had the following conditions: confirmed history of SARS-CoV-2 infection, had received current prophylactic treatment or investigational agents against COVID-19 within 90 days, had unstable underlying diseases that may compromise the immune responses, had a history of vaccine hypersensitivity, were pregnant, were immunocompromised or receiving immunosuppressive agents. Written informed consent was obtained prior to recruitment.

The recruitment was conducted in two phases. The initial phase was an exploratory phase to determine the immunogenicity and safety of 0.1 mL of ChAdOx1 (1/5th of standard intramuscular dose) or 0.05 mL BNT162b2 (1/6th of standard intramuscular dose) as a booster via intradermal injection for each two-dose intramuscular primary series vaccination (prime) group, which included CoronaVac, ChAdOx1, or BNT162b2. There were a total of six groups and the planned sample size was ten per group. The extended phase was aimed to focus on the CoronaVac- and ChAdOx1-prime groups, the main primary series regimens in Thailand, with 35 additional participants enrolled in each group to increase the power of the analysis. However, no additional participants were enrolled for the ChAdOx1 prime- intradermal ChAdOx1 booster group, due to poor immunogenicity data found in the initial phase. The randomization lists were generated before each phase initiation using the Sealed Envelope^TM^, an online software application for the initial phase (Group 1–6) and the extended phase (Group 1–2) (Figure 1). At every study visit, the history of signs, symptoms, and potential exposure to SAR-CoV-2, including any confirmed COVID-19 diagnosis, were reviewed by the research staff. The nasal-swab antigen detection testing was performed using the SD Biosensor Standard Q Covid Ag (SD Biosensor, Inc., Gyeonggi-do, Korea) during the screening visit.

Blood samples were collected for humoral (binding antibody and neutralizing antibody) and cellular (ELISpot) immune response assessment at pre- and 2 weeks post intradermal booster vaccination. The participants were observed for at least 30 min following the vaccination for any immediate adverse events. The participants were instructed to submit self-assessment reports using an electronic diary (eDiary) in Google Form for seven days after each dose of vaccination for any adverse events (AEs), including both solicited local and systemic reactions. The solicited local AEs include pain, erythema, and swelling/induration at the injection site, and localized axillary lymphadenopathy or swelling or tenderness ipsilateral to the injection arm. The systemic AEs include headache, fatigue, myalgia, arthralgia, diarrhea, dizziness, nausea/vomiting, rash, fever, and chills. The severity of solicited AEs was graded using a numerical scale from 1 to 4 based on the Common Terminology Criteria for Adverse Events–Version 5.0 guide by the United States National Cancer Institute (NCI/NIH).

The study protocol was registered at Thai Clinical Trial Registry (TCTR20210907003, TCTR20211102006), and approved by the Siriraj Institutional Review Board (COA no. Si635/2021 and Si894/2021).

### 2.2. Chemiluminescent Microparticle Assay (CMIA) for Anti-SARS-CoV-2 Binding Antibody

The plasma samples were isolated from the blood collected in tubes with sodium citrate solution and stored at −80 °C. The level of antibody (IgG) against the receptor binding domain (RBD) of the SARS-CoV-2 spike protein (Sl subunit) was determined by a CMIA, using the SARS-CoV-2 IgG II Quant (Abbott, List No. 06S60) on the ARCHITECT I System. This assay linearly measures the level of antibody between 21.0 and 40,000.0 arbitrary unit (AU)/mL, which was converted later to the WHO International Standard concentration as the binding antibody unit per mL (BAU/mL), following the equation provided by the manufacturer (BAU/mL = 0.142 × AU/mL).

### 2.3. 50% Plaque Reduction Neutralization Test (PRNT50) against SARS-CoV-2 Strains

The neutralizing antibody titers against delta and omicron (BA.1) strains were determined by the 50% plaque reduction neutralization test (PRNT_50_) on post-booster samples. The method was described previously [18]. The titer of each sample was defined as the reciprocal of the highest test serum dilution; the virus infectivity was reduced by 50% of an average plaque count in the virus control wells and was calculated by using a four-point linear regression method. The PRNT_50_ titers below the positive cutoff of 10 were arbitrarily assigned a value of 5.

### 2.4. IFN-γ ELISpot

Cellular immunity was determined by IFN-γ ELISpot (Mabtech, Nacka Strand, Sweden) to ancestral strains on a subset of participants for each group (*n* = 20). Peripheral blood mononuclear cells (PBMCs) were counted and stimulated with S-peptides that consisted of 100 peptides from spike proteins, and nucleoprotein-membrane protein-open reading frame protein (NMO)-peptide pools, consisting of 101 peptides from nucleocapsid (*n*), membrane (M), open reading frame (ORF) 1, non-structural protein (nsp) 3, ORF-3a, ORF-7a, and ORF8 proteins. Negative controls contained only cell culture media, while positive controls contained an anti-cluster of differentiation 3 (CD3) at a dilution of 1:1000. ELISpot plates were then incubated for 20 h at 37 °C and 5% CO_2_, washed and developed using a conjugated secondary antibody that was bound to membrane-captured IFN-γ. The plates were read using IRIS (Mabtech) and spots were analyzed using Apex software 1.1 (Mabtech) and converted to spot-forming units (SFU) per million cells.

### 2.5. Statistical Analysis

The immunogenicity and reactogenicity of the intradermal booster in the CoronaVac-prime and ChAdOx1-prime groups were compared with a previous study of the standard intramuscular booster injection of similar regimens and timing of booster vaccinations [19]. The sample size of 45 in each group included in the extended phase provides the 90% power to determine the non-inferiority between the intradermal and intramuscular route if the lower bound of the 95% confidence interval (CI) around the ratio of anti-RBD SARS-CoV-2 IgG antibody geometric mean concentrations (GMC) two weeks after vaccine injection is at least 0.67, as recommended by WHO (https://www.who.int/publications/m/item/WHO-TRS-1004-web-annex-9, access date 3 August 2021). The AE endpoints were presented as frequencies and the Chi-square test was used to test for statistical difference. The immunological endpoints of anti-SARS-CoV-2 RBD IgG and PRNT_50_ titer were reported as GMC and geometric mean titers (GMT) with 95% confidence interval (CI), respectively. Unpaired t-tests were used to compare IgG GMCs and IFN-γ SFU between groups. Pearson’s correlation coefficient was used to assess the correlation between Log10 of anti-SARS-CoV-2 RBD IgG and Log10 of PRNT_50_. All statistical analyses were performed using the GraphPad Prism 9 version 9.2.0 (283) (GraphPad Software, San Diego, CA, USA), except for the ANOVA analysis of the anti-SARS-CoV-2 RBD IgG among different age groups that was performed using STATA version 17 (StataCorp, LP, College Station, TX, USA).

## 3. Results

### 3.1. Baseline Characteristics of Study Participants

The baseline characteristics of the cohort and results of the initial phase are shown in Appendix A. For the main analysis in the extended phase, a total of 45 participants in the CoronaVac prime-intradermal ChAdOx1 boost (CoronaVac-ChAdOx1), CoronaVac prime-intradermal BNT162b2 boost (CoronaVac-BNT162b2) and ChAdOx1 prime-intradermal ChAdOx1 boost (ChAdOx1-BNT162b2) groups were included, and their baseline demographic characteristics are shown in Table 1. The overall median age (interquartile range: IQR) was 39 years (30–46), 43.6% were male and the BMI was 24.0 kg/m^2^. The median (IQR) interval between the second dose of primary series and the ID booster vaccination was shorter in the ChAdOx1-BNT162b2, but was not statistically significant compared to the CoronaVac-primed groups. Consequently, the ChAdOx1-BNT162b2 group also had a higher baseline antibody concentration than CoronaVac-primed groups.

### 3.2. Humoral Immune Responses of Intradermal ChAdOx1 and BNT162b2 Booster

Two weeks following the intradermal booster, the anti-RBD IgG GMC significantly increased in all three groups (*p* < 0.0001). The CoronaVac-BNT162b2 group had significantly higher GMCs than the CoronaVac-ChAdOx1 and ChAdOx1-BNT162b2 groups. The ratio between post-vaccination and pre-vaccination for the CoronaVac-ChAdOx1, CoronaVac-BNT162b2 and ChAdOx1-BNT162b2 groups were 48.2 (33.4–69.6), 57.8 (42.5–78.4) and 7.6 (6.3–9.0), respectively (Figure 2 and Appendix A). Importantly, compared to their respective intramuscular heterologous vaccine regimens previously reported [19], there was no statistically significant difference in GMC for the CoronaVac-ChAdOx1 group, while the GMC of intradermal boosting in both CoronaVac-BNT162b2 and ChAdOx1-BNT162b2 groups were around 2.6 and 1.8-fold lower than the intramuscular boosting (*p* < 0.001).

Following the intradermal booster dose, the neutralizing antibody titers against the delta variant in all three groups were marginally lower (1.3–1.6-fold) than the intramuscular booster of the same regimens, although the titers for ChAdOx-BNT162b2 were significantly lower for intradermal vaccination, compared with the titers for intramuscular vaccination (Figure 3A and Appendix A). Interestingly, compared to their respective intramuscular vaccine regimens, the neutralizing antibody titers against the omicron variant were significantly lower (5.4–5.8-fold) with intradermal BNT162b2, but were similar with intradermal ChAdOx1 (Figure 3B and Appendix A).

### 3.3. Cellular Immune Responses of Intradermal ChAdOx1 and BNT162b2 Booster

An intradermal booster increased the T cell responses against the spike protein for all three groups, with significantly higher spike-specific T cell responses induced by the intradermal BNT162b2 booster compared with the intradermal ChAdOx1 booster (Figure 4A, and Appendix A). For T cell responses against the NMO proteins, no statistically significant differences were observed for any of the groups following the intradermal booster (Figure 4B and Appendix A).

### 3.4. Reactogenicity of Intradermal Booster Injections

Local injection site reactions were similarly common among the groups and the proportions were not different compared with the reactogenicity of intramuscular injection of the similar regimens (Figure 5A). However, the severity of reactions for the intradermal injection was milder compared to intramuscular injection. The systemic adverse effects were significantly less frequent and milder in intradermal compared to intramuscular booster injections (Figure 5B).

## 4. Discussion

This study found that the administration of fractional ChAdOx1 (1/5th of standard dose) and BNT162b2 (1/6th of the standard dose) as a booster via intradermal injection to individuals previously vaccinated with two-dose CoronaVac or ChAdOx1 standard primary series generated robust immune responses against the delta and omicron variant. Importantly, similar immunogenicity measured by neutralizing antibodies against omicron was generated following the intradermal booster, compared to the standard intramuscular ChAdOx1 booster to CoronaVac-prime participants. However, lower antibody responses were found for the intradermal administration of fractional dosages of BNT162b2, compared with the corresponding vaccine regimen given via intramuscular injection; their antibody concentrations were similar to the CoronaVac-ChAdOx1 group. This is the first study to report the immunogenicity of intradermal fractional dosing against the omicron variant. Our findings have important implications on the use of COVID-19 vaccines in Thailand and in settings where COVID-19 vaccines are in limited supply. It is important to note that this study was conducted when there was a major shortage of COVID-19 vaccines. For future vaccines, there may be a similar situation and novel strategies, such as intradermal vaccination, may be needed.

Intradermal vaccination is thought to improve global access to COVID-19 because of the use of smaller doses, provided it offers the same immunogenicity and protection [20]. However, to date, there are only a few studies that have evaluated this strategy for COVID-19 vaccines. The initial phase of the study revealed some important findings of each vaccine when administering intradermally following various priming vaccinations. This includes the concentration of antibodies at the time of the booster dose, affecting the booster responses. Another observation is that while participants vaccinated with CoronaVac had much lower IgG antibodies after 12 weeks, the strong anamnestic response following the booster dose suggests that sufficient immune memory was induced. This was substantiated in the extended phase. While neutralizing antibodies are thought to be the primary mechanism of protection [4], there is currently no identified level for protection.

Our findings indicate that robust immunogenicity is generated following an intradermal booster dose of ChAdOx1 given at 1/5th of the standard dose in participants who previously received the standard intramuscular Coronavac, which is consistent with the findings from two other studies. These two studies found similar and non-inferior humoral and cellular immune responses between the standard intramuscular ChAdOx1 booster dose and the intradermal ChAdOx1 booster dose at 1/5th of standard dosing [21,22]. For intradermal boosting using BNT162b2, our study findings were also consistent with a previous study that reported significantly lower anti-RBD IgG concentrations and neutralizing antibodies against the delta variant, when compared with corresponding intramuscular injections in individuals vaccinated with the CoronaVac primary series [23]. This is despite the use of different vaccine dosages for intradermal injection (1/3rd vs. 1/6th of standard dose). Of note, none of these studies examined neutralizing antibodies against omicron. The neutralizing antibodies titers against omicron found in our study following intradermal boosting of fractional doses were similar across the three groups, but the titers in each group were different, when compared with their corresponding vaccine regimen given via intramuscular injection. High vaccine effectiveness (>93%) against symptomatic COVID-19 patients, COVID-19-related hospitalization, ICU admission, and death have been reported for individuals who were vaccinated with CoronaVac primary series and boosted with ChAdOx1 [24]. However, this study was conducted pre-omicron. As recent circulating omicron subvariants (BA.4, BA.5) were found to escape immunity from standard intramuscular immunization, the lower neutralizing titers via the intradermal route would probably not be clinically significant.

We found that intradermal boosting with BNT162b2 induced higher T-cell responses to spike proteins than ChAdOx1 among individuals previously vaccinated with CoronaVac. However, we were not able to directly compare the T cell responses between those induced by intradermal injection and intramuscular injection. Previous studies of the intradermal fractional ChAdOx1 booster in individuals who received CoronaVac primary series found similar T cell responses to the standard intramuscular ChAdOx1 booster at two weeks post-booster dose; these responses, however, were lower in patients who were given an intradermal injection than intramuscular injection at four weeks post-booster dose [21,22]. For BNT162b2, the T cell responses were lower following the intradermal fractional dose than the standard intramuscular dose at two weeks [23]. The clinical relevance of the lower T cell responses through intradermal injection is unclear but may be due to the lower vaccine dosage. Further studies, including dose escalation, that titrate the vaccine dosage to reach the expected immunity levels, but within acceptable adverse reactions, may be needed. This may also be relevant for new COVID-19 vaccines, such as the bivalent Wuhan- and omicron-variant vaccine and omicron-specific mRNA vaccines that are currently being investigated.

Our findings of lower and milder systemic reactions are consistent with those that were reported for intradermal injections of COVID-19 vaccines and other vaccines [14,21,22]. Intradermal administration may, therefore, increase vaccine uptake among those hesitant about vaccination due to the high reactogenicity concern associated with the standard intramuscular injection of some COVID-19 vaccines. However, intradermal injection would require specific training on administration and supply of special syringes, which have implications on implementation programs.

Our study has some limitations. Although the RT-PCR and anti-nucleoprotein tests were not performed to rule out natural infection during the study, the participants were screened by history of illness and exposure to COVID-19, as well as antigen detection testing, and were closely observed during the 2-week study period. Therefore, they were less likely to have unrecognized infections. Our sample size is small, and we did not have a direct comparison group in the same cohort with intramuscular injection; therefore, the results should be interpreted with caution. Another limitation is that our results on the ChAdOx1 and BNT162b2 fractional intradermal booster may not be generalizable to other COVID-19 vaccines and populations who have received other COVID-19 vaccines as primary series.

## 5. Conclusions

In conclusion, we found that intradermal injection of fractional BNT162b2 following CoronaVac- and ChAdOx1-prime generate robust antibody responses against SARS-CoV-2 delta and omicron variants, but at a lower concentration than the intramuscular route. The lower BNT162b2 response following ID booster vaccination might also be related to the lower antigen dose, but whether this is associated with lower efficacy needs to be studied further. On the other hand, the intradermal fractional ChAdOx1 booster induced similar antibody responses to the intramuscular route following CoronaVac-prime. The intradermal route had lower and milder systemic adverse reactions. Further studies are needed to evaluate the clinical relevance of these findings and the feasibility of administration of intradermal COVID-19 vaccines. Our study has significant implications for Thailand and other similar settings, where CoronaVac and ChAdOx1 vaccines were given as primary series and have limited access to mRNA vaccines.

## Figures and Tables

**Figure 1 vaccines-10-01497-f001:**
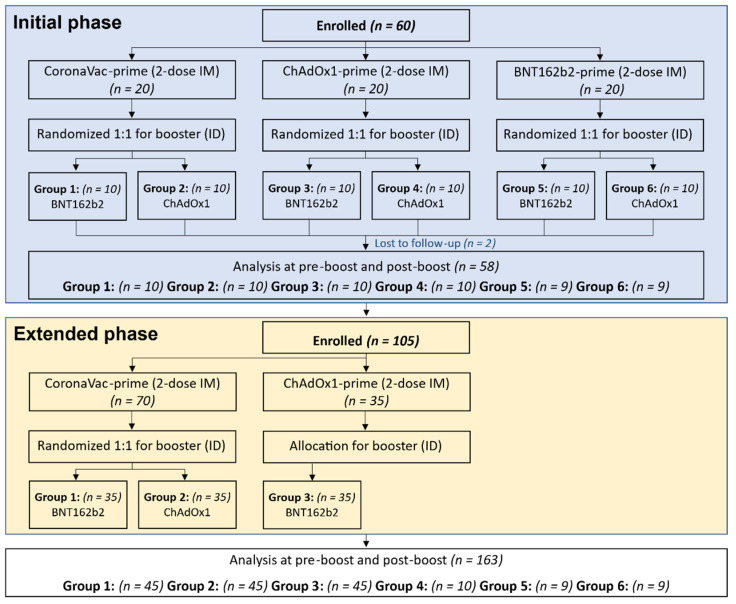
Consort Diagram showing enrollment of each group in initial phase and extended phase. Group 1 and 2 received two doses of CoronaVac intramuscularly (CoronaVac-prime), Group 3 and 4 received two doses of ChAdOx1 intramuscularly (ChAdOx1-prime), and Group 5 and 6 received two doses of BNT162b2 intramuscularly (BNT162b2-prime). Each group was randomized to receive intradermal BNT162b2 (0.05 mL, 1/6th of standard dosage) or ChAdOx1 (0.1 mL, 1/5th of standard dosage). Additional participants were enrolled in the extended phase to join Groups 1, 2, and 3.

**Figure 2 vaccines-10-01497-f002:**
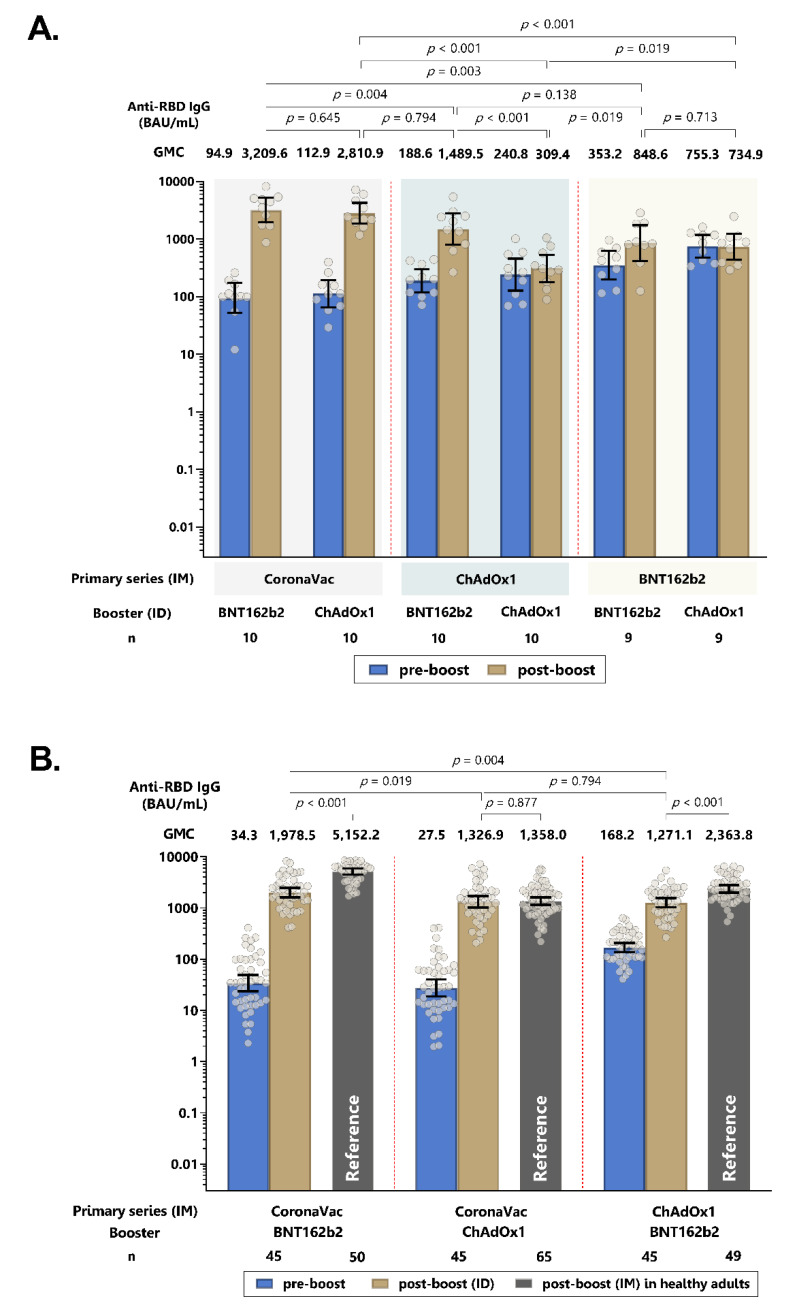
Geometric mean concentration of anti-RBD IgG pre- and post-intradermal (ID) booster dose in the initial phase (**A**) and in initial and extended phase (**B**). Intramuscular (IM) boosting of the same vaccine regimens from the previous study [19] is included as a reference group. Error bars represent geometric mean concentration, plus 95% confidence intervals. BAU: binding antibody units.

**Figure 3 vaccines-10-01497-f003:**
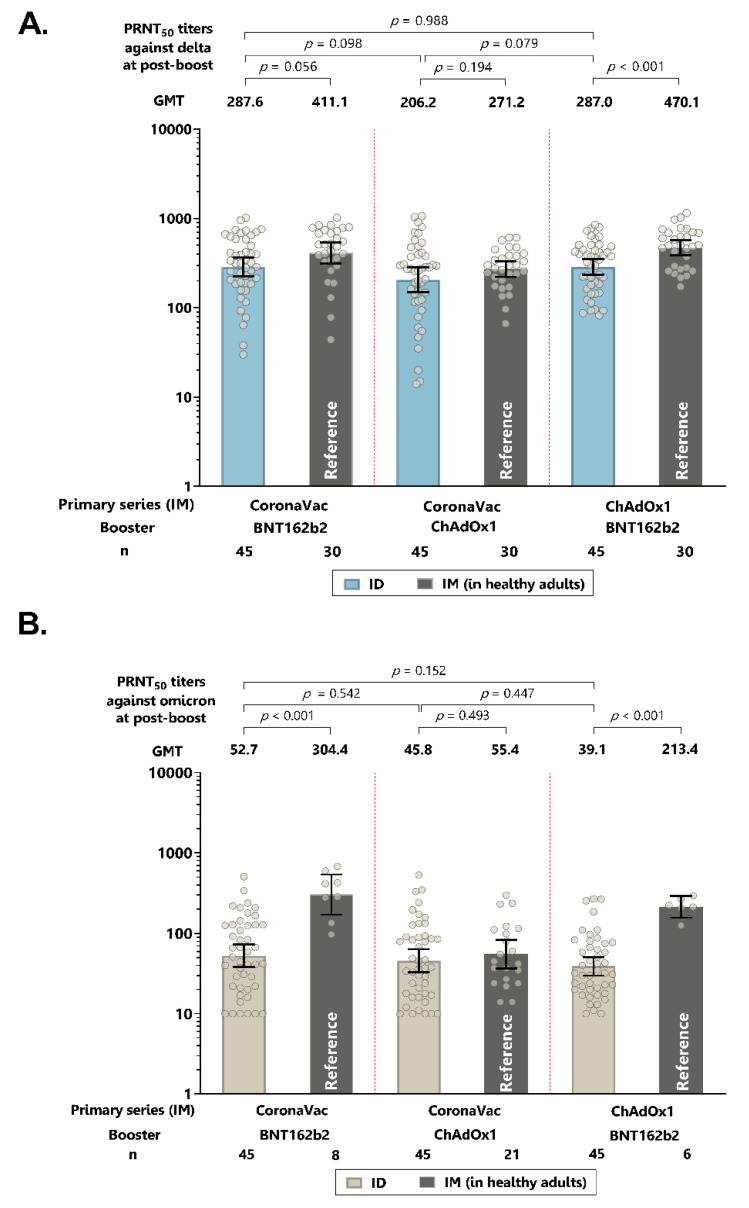
Neutralizing antibody titers of live virus plaque reduction neutralization assays (PRNT_50_) against delta strain (**A**) and omicron strains (**B**) following intradermal (ID) booster. Intramuscular (IM) boosting of the similar vaccine regimens from the previous study [19] is included as the reference group. Error bars represent geometric mean titers plus 95% confidence intervals.

**Figure 4 vaccines-10-01497-f004:**
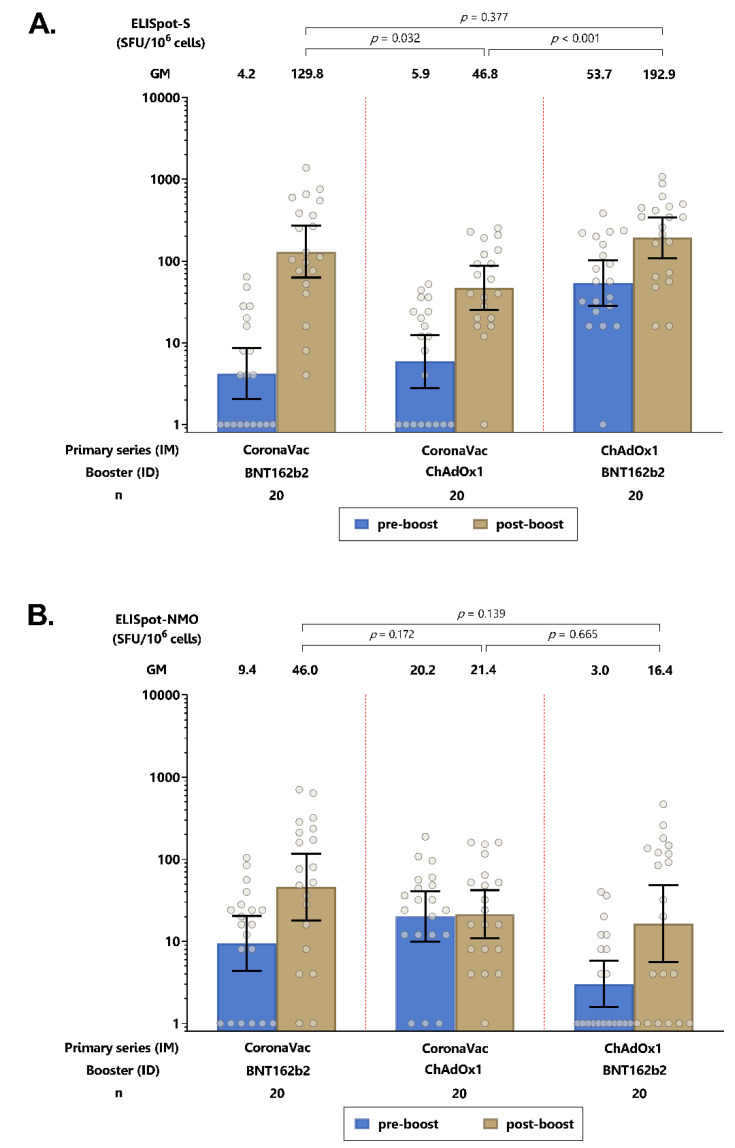
SARS-CoV-2 antigen-specific T-cell response by ELISPOT after intradermal (ID) booster. (**A**) Spike-specific T cell responses. (**B**) Nucleocapsid-membrane-open reading frame-T cell responses. Error bars represent geometric mean concentrations plus 95% confidence intervals. SFU: spot-forming units.

**Figure 5 vaccines-10-01497-f005:**
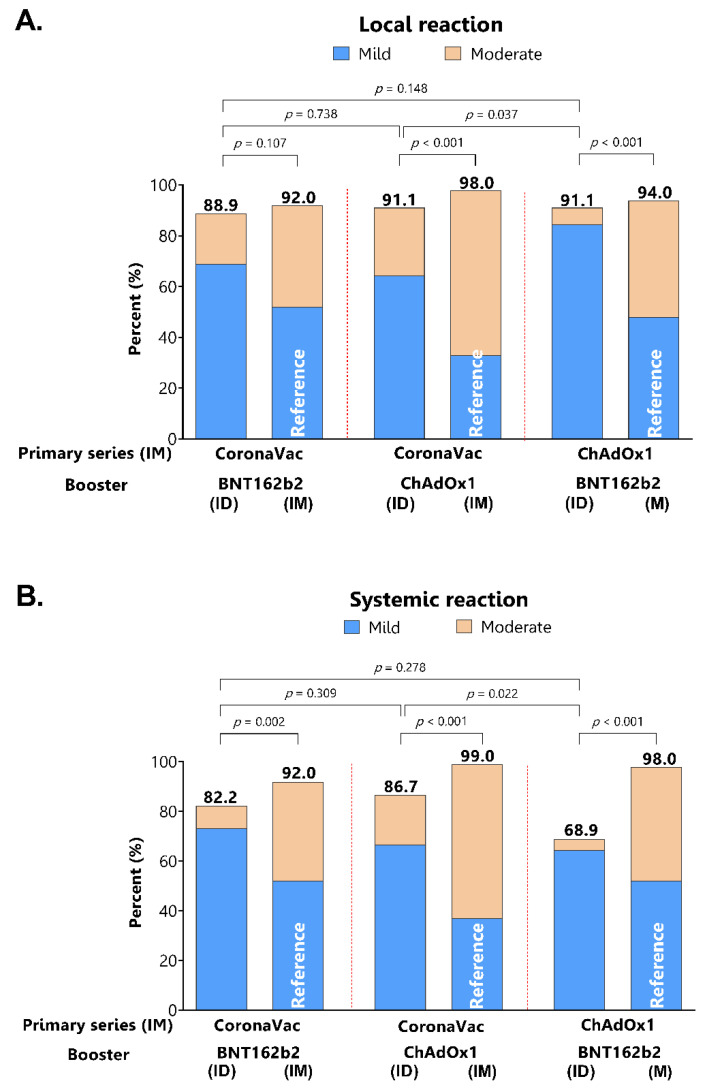
Self-reported adverse events following intradermal boosting consist of local injection site reaction (A) and systemic reaction (B). Intramuscular boosting of the similar vaccine regimens from the previous study [19] is included as a reference group.

**Table 1 vaccines-10-01497-t001:** Baseline characteristics of the participants of each group in the initial and extended phase.

Primary Series (IM-IM)-Booster (ID)	Types of Vaccines				
	All	CoronaVac- BNT162b2	CoronaVac- ChAdOx1	ChAdOx1-BNT162b2	*p*-Value
Number of subjects, *n* (%)	135 (100.0)	45 (33.3)	45 (33.3)	45 (33.3)	-
Age (years), median (IQR)	39.0 (30.0, 46.0)	38.0 (29.0, 44.0)	39.0 (29.0, 45.0)	43.0 (30.0, 50.0)	0.186
Male, *n* (%)	60 (44.4)	18 (30.0)	23 (38.3)	19 (31.7)	0.533
Body mass index: BMI (kg/m^2^), median (IQR)	24.0 (21.5, 26.6)	24.4 (22.3, 26.8)	23.6 (20.4, 26.5)	23.9 (21.5, 26.4)	0.719
Interval between last dose of primary series and booster (weeks), median (IQR)	9.9 (7.4, 12.0)	11.6 (8.9, 16.3)	10.3 (9.1, 12.6)	7.4 (6.0, 9.6)	0.083

## Data Availability

The data presented in this study are available upon request from the corresponding author.

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
