# Peer review of "Evaluation of the Safety and Immunogenicity of Fractional Intradermal COVID-19 Vaccines as a Booster: A Pilot Study"

_vaccines, 2022, doi:10.3390/vaccines10091497_

Round 1
Reviewer 1 Report
Thank you for the opportunity to review a very relevant and interesting paper. There are a few areas for improvement.
1) Introduction should include references for heterologous boosting for COVID-19, for example, would consider: Liu et al in Lancet (Com-COV); Stuart et al in Lancet (Com-COV2), Nordstrom et al in Lancet Reg Health, Atmar et al in NEJM, Tenbusch in Lancet ID, Costa Clemens in Lancet.
2) In the sentence which includes reference 9, the sentence should mention that the vaccine was for influenza.
3) Methods do not mention randomization methods, or blinding/masking. Who was blinded? Were clinicians and participants and laboratory blinded to treatment assignment? Or was it open-label?
4) There is no mention of whether any COVID infections or COVID testing happened during the study and whether participants who acquired COVID during participation were still included in the immunogenicity analyses.
5) In Figure 1, in the extended phase, Group 3 is indicated as randomized, but it seems that it was not randomized. All ChAdOx1 primed participants were assigned to receive BNT162b2. Group 3 should have a separate box for treatment assignment because it was not randomized along with the CoronaVax primed participants.
6) Much is made about the higher Neutralizing antibody titer in the ID Coronavax-ChAdOx1 group compared to IM but this needs more discussion because it is unexpected and unexplained. The titers are within an order of magnitude of each other, and could the supposed superiority be because the IM Coronavax-ChAdOx1 responses were measured as unusually low in that cohort?
7) One conclusion is about lower antibody responses to BNT162b2 ID boost but that could be the effect of underdosing, and while the differences were statistically significant, they may not be clinically significant in terms of efficacy. These caveats should be discussed.
8) "Whether the immunogenicity induced by intradermal vaccination observed in our study is sufficient to protect against circulating omicron remains to be determined." This could be updated. As it turned out, no IM vaccine protected very well against Omicron infection, so the lower neutralizing titers from intradermal vaccination would probably not be clinically significant.
9) In the conclusion, there appears to be an error in the second sentence. "On the other hand, intradermal fractional ChAdOx1 booster induced similar antibody responses to intradermal route following CoronaVac-prime." Should the second intradermal be changed to intramuscular?
10) Based on the results (including the early phase) another conclusion might be that homologous and heterologous boosts may have unpredictable immunogenicity results--not all are beneficial-- and need to be tested. Another conclusion might be that the decreased immunogenicity of some vaccines could be remedied by an appropriate heterologous boost.
11) The English in the discussion has a few awkward areas. It also ends with an incomplete sentence at the end: "We also"
Author Response
Thank you for your comments. Please see the attached file for our response.

Reviewer 2 Report
Niyomnaithan et al conducted a single-center trial that evaluates the immunogenicity of fractional ID boosting with the BNT162b2 or Chadox1 COVID-19 vaccine in healthy Thai adults. In the initial phase of the trial they assessed immunogenicity in six groups consisting of 10 participants who received either Pfz, AZ or CoronaVac (all administered intramuscularly) as a primary series. In the extended phase of the trial, they continue with the groups that received CV or Chd1 as a primary series and recruit 35 more participants in these groups. The Chd1-Chd1 group is dropped due to poor results in the initial phase. In the extended phase, historical data from a trial that started three months prior to this trial are used as a reference. The authors conclude that the groups (CV and Chd1 prime series) that received the fractional intradermal booster with BNT have robust immune responses, albeit lower than the references. The group boosted with fractional intradermal Chd1 (CV prime series) has better immune responses than the reference group.
General comment:
These are novel results that are well presented in this manuscript. The results are of interest to a broad public, and more specifically in the Thai setting or in other countries that have primarily been vaccinated with Chd1 and CV and that have limited access to mRNA COVID-19 vaccines. The design of the study is sound, but would have been better if the control group was part of the study itself, and randomisation would have included both intervention (ID) and control (IM) groups. The reference group that is used, should be described more clearly in the manuscript (e.g. demographics, time interval between primary and booster series, serological tests performed). Ideally, T-cell responses of the reference group would be assessed and included as well.
Secondly, in analysing booster responses a pre-post booster fold change in antibody concentration / titer is more appropriate (in contrast to absolute concentrations / titers) are pre-booster levels vary. Please add these to the results.
Specific comments:
* Please elaborate on why a 1/6th dose was chosen for the BNT vaccine and a 1/5th dose for Chd1.
* The in/exclusion criteria state that a confirmed history on COVID-19 was an exclusion criterion. Was there any serology testing performed to conform this COVID-19 naive status of participants? Were PCR tests performed during the trial? Was SARS-CoV-2 anti-N performed? Are there any chances the study contains persons who had asymptomatic disease? And what about the controls?
*Please describe the blinding of the trial in the method section (open-label?)
*The result section states that the period between the last vaccination of the prime series and the booster of the BNT group was shortest. Please include these results in Table 1
*Can the baseline characteristics of the reference groups be added to Table 1 (including time between last dose of primary series and booster)?
*Some brackets and spaces are missing in Table 1.
*Please include figure descriptions in the supplements.
*Second alinea of the discussion states that ID vaccination is touted to improve vaccine access, giving ref 14 and 15 as sources. However, ref 15 is quite critical on fractional dosing, is this reference correct?
*Please change ref 13 to the most recent version of the MedRxiv article.
*Consider the following article on ID vaccination that has not been incorporated in the manuscript:
https://www.medrxiv.org/content/10.1101/2021.07.27.21261116v1
*the last sentence of the discussion is not finished
Author Response

(The authors gave the same response as above.)

Reviewer 3 Report
As stated by the authors, this study is a preliminary one with some limitations. Nevertheless, the data reported, which are interesting, are mandatory to pave the way of a more robust and extensive study which should be undertaken.
Author Response
We are really appreciated with your comments. Thank you.